# Trace Metals (As, Cd, Cr, Cu, Hg, Ni, Pb, Zn) and Stable Isotope Ratios (δ^13^C and δ^15^N) in Fish from Wulungu Lake, Xinjiang, China

**DOI:** 10.3390/ijerph18179007

**Published:** 2021-08-26

**Authors:** Fengyu Liu, Min Li, Jianjiang Lu, Zhiqing Lai, Yanbin Tong, Manli Wang

**Affiliations:** 1Key Laboratory of Environmental Monitoring and Pollutant Control of Xinjiang Bingtuan, School of Chemistry and Chemistry and Chemical Engineering, Shihezi University, Shihezi 832000, China; liufengyu_hj@163.com (F.L.); tongyanbin@sina.com (Y.T.); wangmanli8090@163.com (M.W.); 2The Key Laboratory of Sea Floor Resource and Exploration Technique, Ministry of Education College of Marine Geoscience, Ocean University of China, Qingdao 266100, China; zqlai@ouc.edu.cn

**Keywords:** trace metals, stable isotope ratios, Wulungu Lake, fish, risk assessment

## Abstract

Wulungu Lake is a vital fishery area in Xinjiang. However, the concentration, enrichment rules, and sources of As, Cd, Cr, Cu, Hg, Ni, Pb, and Zn in the aquatic organisms, have rarely been investigated. The results suggest that the concentrations of As, Ni, Pb, and Zn were higher than those recommended by the national standards for edible fish in some species. Hg, Ni, Cu, Pb, Cr, and Zn in the fish were dependent on the concentration of trace metals in the water environment (*p* < 0.05). The body weights of the fish were significantly negatively correlated with only Hg (*p* < 0.05); however, their body lengths were significantly positively correlated with As, Cu, Zn, and Hg contents. Values of δ^13^C ‰ (δ^15^N ‰) for the entire fish food web was found to range from −19.9‰ (7.37‰) to −27.7‰ (15.9‰), indicating a wide range of trophic positions and energy sources. Based on the linear correlation, As, Cu, Cd, and Zn contents were positively correlated with δ^15^N (*p* < 0.05), and bioaccumulation was observed in the fish. The target hazard quotient (*THQ*) of all fish species was less than 1, indicating the absence of potential risks to human health.

## 1. Introduction

Xinjiang is representative of one of the most arid and semi-arid regions in China [1]. Owing to the industrial development in Xinjiang, especially that of chemical industries [2], the surface water quality in the region has considerably deteriorated. Water quality deterioration is manifested in terms of trace metal pollution in the Yili River basin [3], with 114 lakes across central and eastern Tianshan exhibiting an increased concentration of trace metals in surface waters; in some lakes, the surface water area constituting trace metals exceeds 1 km^2^. The total surface water area comprising of trace metals spans approximately 6400 km^2^, which is 7.7% of the total lake area in China [3]. The Yili River basin is the source of mineral resources; thus, heavy industries such as mining and smelting are located in the basin around the lake area. Densely populated areas, and correspondingly high anthropogenic activities, in the vicinity of the lakes, result in more severe trace metal pollution in the lake sediments. In contrast, lakes located far away from human settlements are subjected to less point-source and non-point-source pollution, and thus exhibit decreased trace metal concentrations and relatively good water quality [4]. Since 1969, the Erqisi River has introduced 1.83 × 10^8^ m^3^ of water into Wulungu Lake annually. The Erqisi river flows through the main metal metallogenic belt in Xinjiang, China, which may bring some trace metal pollutants into Wulungu Lake [5]. The average contents of heavy metals Cu, Zn, As and Pb in the soil of the Erqisi River Basin were 52.827, 471.320, 11.697 and 46.533 mg/kg, which were higher than the background values of heavy metals in the soil of Xinjiang. The Erqisi River may carry these heavy metals into the Wulungu Lake [6]. Huang [7] concluded that the average contents of Cu, Zn, Cr, Ni and Pb in the surface sediments of Wulungu Lake are 11.88, 21.12, 9.44, 18.83, 14.20 µg/g, and Wulungu Lake was slightly polluted. Cd is the main pollution element with strong potential harm. Wulungu Lake is a substantial fishing base in Xinjiang and is known as the “Northern Country Fishing Township,” accounting for more than 1/3 of the total annual fishery production in Xinjiang [8], thereby the ecological problem of Wulungu Lake caused by the trace metals pollution of anthropogenic activities has attracted much attention.

Trace metal pollution is widely present in the atmosphere, water, soil, and solids. It is primarily caused by mining, waste gas emissions, irrigation using sewage water, and when products that exceed the trace metal concentration standards are used. Bioaccumulation in aquatic environments is an area of increasing concern. Chowdhury et al. [9] found in adult rainbow trout that dietary Cd resulted in a 26-fold increase in plasma Cd level over 45 days. Despite low concentrations of trace metals in water and sediment, the characteristics of bioaccumulation in organism tissue can be transferred to fish tissue (muscle, gills, liver, stomach, and intestine), they may ultimately accumulate to dangerous levels and pose a potential risk to human health [10,11,12].

Fish that have been exposed to sewage will ingest toxins; some chemical toxins can be stored in their muscles, fats, and viscera for an extended period. The content and toxicity of trace metals in muscles and organ tissues of different fish species differ [13,14,15,16]. Freshwater organisms have the highest sensitivity to Cu and the lowest sensitivity to Zn [7]. The increase in trace metal contents in daily living environments is a direct result of human activities [17]. Once the trace metal content exceeds the threshold, it directly affects human health, endangers life, and leads to serious environmental degradation. 

Approximately 90% of the total human intake of trace metals is through their diet [18]. Among various foods, fish are widely consumed because of their high nutritional value and are regarded as the most suitable biological indicators in aquatic ecosystems. Sallam et al. [19] conducted a health risk assessment of trace metals in some fish in Lake Manzala, Egypt; the results showed that these fish posed health risks to consumers. Moreover, Liu et al. [20] found that the fish in the mangrove wetland of Qiao Island in Zhuhai was heavily polluted by Cr. The health risk assessment results showed that the consumption of commercial fish acquired from this area poses a considerable health risk, and should be taken seriously. Yan et al. [21] used the target hazard quotient (*THQ*) method recommended by USEPA (2009) to estimate the health risks of consuming karst cavefish containing trace metal. The results showed that although cavefish consumption does not pose health risks due to trace metals, Cu and As concentrations in the fish are close to their respective risk limits; hence, their presence in the cavefish should be regarded as a matter of concern. The food chain transmission of trace metals can be traced back to the 1960s, after the first water-borne decline in Japan attracted attention toward the transmission of trace metals (especially Hg) in the food chain. Bioaccumulation of trace metals in organisms poses serious risks to their health and well-being. The Hg content in muscle and liver tissues of fish and birds in the northern waters of Baffin Bay has been observed to be directly related to the biological trophic level [22]. Du et al. [23] showed that the transport of Cr, Co, Ni, Zn, Cd, and Pb along the food chain in the summer and winter decreased significantly in Daya Bay. In the fish-heron food chain in the Sanjiang Plain, Hg content magnified with an increase in the trophic level [24].

Some scholars have investigated the concentration, spatio-temporal distribution, sources, and potential ecological risks of trace metals in lakes; however, there are only a few reports on the ecological environmental influences of trace metals in the lakes of Xinjiang. In particular, there is a lack of research on the correlation between trace metal pollution in lake water, sediment, and fish. Furthermore, there are few reports on the accumulation and transfer of trace metals in the nutrient base and the health risk assessment of trace metals using carbon and nitrogen stable isotopes. Only by systematically studying the content, classification, sources, and enrichment of trace metals in lake water, sediment, and fish at different trophic levels can we comprehensively grasp the characteristics of pollution caused by trace metals in these entities and analyze them scientifically. 

In this study, the concentrations of As, Cd, Cr, Cu, Hg, Ni, Pb, Zn in 10 species of common edible fish, water and sediments, respectively, collected from Wulungu Lake were investigated. Samples were analyzed to find the existence of As, Cd, Cr, Cu, Hg, Ni, Pb, and Zn contamination in fish from this area, and the possible contamination sources if existed. Data were processed to investigate whether there were significant differences in eight trace metals among different fish species, feeding habits, to further reveal the factors that influence metal concentrations. The relationship between trace metals content in water environment and fish trace metals content, the relationships of eight trace metals to δ^15^N and δ^13^C, as well as fish weight and length were also studied to illustrate specific biomagnification and bioaccumulation processes, respectively. The most common edible 10 species fish (which come from four different food habits) from Wulungu Lake were used as research objects. The linear changes in the 10 above-mentioned species of fish in ratios of stable isotopes of carbon (δ^13^C) and nitrogen (δ^15^N) [25], can be employed as a bioindicator for trace metals presence and pollution in lakes in arid regions. Furthermore, this study shed light on the health risk of consuming fish based on daily intake calculation.

## 2. Materials and Methods

### 2.1. Study Area

Wulungu Lake is located in Fuhai at the Altai region of Xinjiang, China (87.18° E–87.71° E, 46.76° N–47.33° N) (Figure 1). It is approximately 30 km wide in the north–south direction, 35 km wide in the east–west direction, with an area of 827 km^2^. The lake is filled with 1.83 × 10^8^ m^3^ of water per year by the Erqisi River. It is the second-largest lake in Xinjiang and is an important fishing ground. The food resources from the lake, especially the fish, are a source of high-quality protein for the people of Xinjiang. Today, the economic development of the Wulungu Lake basin and the increase in population has resulted in a large amount of wastewater discharge, including agricultural wastewater, domestic sewage, and industrial pollution wastewater, thereby exceeding the self-purification capacity of Wulungu Lake [26].

### 2.2. Sample Collection and Preparation

In May 2019, water samples and sediments were collected at 13 sampling points in Wulungu Lake. Water samples were collected at a depth of about 50 cm from the lake water surface with polytetrafluoroethylene (PTFE) bottles; the sampling time and location were noted on the bottles. All water samples collection procedures were carried out in accordance with Chinese standards [27]. The sediment samples were collected from the same site (the water sample collection place) by Van Veen grab, subsequently plant rhizomes, stones, and other sundries were removed, and finally placed in clean polyethylene bags on which the sample collection time and location were recorded. These were then sealed and frozen for storage at −20 °C. All sediment samples collection, storage and transportation procedures were carried out in accordance with Chinese standards [28]. A total of 118 fish were caught at 3 fishing locations (Y1, Y2, Y3) (every 2 to 5 km) in Wulungu Lake, which belonged to 10 species from four different food habits. All fish are be confirmed healthy according to external physical examination of any signs of abnormality or diseased infected tissue (Table 1). The number, weight, body length, feeding habits, and fishing location of each fish were recorded. The fish were caught by professional fishermen in Wulungu Lake using gillnets and nets. After the fish were collected, they were immediately placed in polyethylene bags and stored at low temperatures (−20 °C). After the samples were thawed and cleaned with pure water, the muscle near the first dorsal fin was extracted for the study, which was washed with deionized water and vacuum dried. For small fish, mixed sampling was used, and evenly ground and loaded samples were taken.

### 2.3. Trace Metal Analysis (Cr, Cu, Cd, Zn, Ni, As, Hg, Pb)

The water samples were filtered using a 0.4 μm filter membrane to remove suspended solids and digested with 10 mL concentrated nitric acid. The water sample volume was made up to 100 mL using a volumetric flask; the sample was transferred into a 25 mL centrifuge tube and analyzed using a plasma mass spectrometer (ICP-MS US PE NexION 2000B, PerkinElmer Waltham, Massachusetts US) [29]. The microwave digestion method with nitric acid and hydrogen peroxide was used to treat the sediment samples and fish. The content of trace metals (Cr, Cu, Cd, Zn, Ni, Pb) was detected by an inductively-coupled plasma mass spectrometer (ICP-MS US PE NexION 2000B, PerkinElmer Waltham, Massachusetts US), according to the previously reported method [29]. For As and Hg analysis, 0.3 g of the dried and homogenized sample (for each element) was digested by heating with mixed acids (HNO_3_:HCl:H_2_O = 1:3:4). After cooling, 2 mL supernatant and 10 mL of diluted HCL (HCL:H_2_O = 1:1) were mixed, and a solid mixture reducing agent (5 g thiourea and 5 g ascorbic acid dissolved in 100 mL water) was added. Furthermore, the resulting solution was diluted with Milli-Q water to 100 mL before its analysis by an atomic fluorescence spectrometer (Jitian AFS-8520, Jitian, Beijing, China). Blank and control experiments were performed for each batch of experiments to eliminate the biases due to the pretreatment of samples. The method has high spiked recovery and good precision (Appendix A).

### 2.4. Stable Isotope Analysis

The fish samples were dried to a constant weight at 65 °C [30], and ground into a fine powder. The carbon and nitrogen stable isotope ratio of the samples (‰) was measured using a stable gas isotope ratio mass spectrometer (Flash EA1112 HT-Delta V Advantage, Thermo Waltham, Massachusetts US).
δ^15^N or δ^13^C (‰) = (R_sample_/R_standard_ – 1) × 1000(1)
where R is the ratio of ^15^N/^14^N or ^13^C/^12^C, and the background materials of δ^15^N and δ^13^C are N_2_ in the atmosphere and carbon in Vienna Pee Dee Belemnite (VPDB). The precision of the analytical technique was <0.20‰ for δ^13^C and <0.30‰ for δ^15^N.

### 2.5. Health Risk Assessment

The *THQ* method introduced by the USEPA was used to evaluate the health risks caused by the consumption of trace metal-containing fish, as per the following formula:(2)$$EDI\;=\;\left(C\times DC\right)/W$$
(3)THQ=EDI/RfD

In Equation (2), *EDI* is the estimated daily intake calculation in μg/(kg·d); *C* is the mean contaminant concentration in μg/g; DC is the aquatic product intake (7.1 g/d) [31]; *W* is the average body weight (male: 66.2 kg, female: 57.3 kg) [32], and *RfD* is the reference exposure dose. The *RfD* values of Cd, Cu, Pb, Cr, Zn, Ni, As, and Hg are 1, 1, 4, 1500, 2, 300, 0.3, and 0.1 μg/(kg·d), respectively [33,34].

The effects of trace metals on human health are not taken into account if the *THQ* value is <1, while *THQ* > 1 is indicative of health risk to humans [35]; the higher the value, the greater the risk. The total risk index is the summation of individual trace metal risk indexes [36]. 

### 2.6. The Average Pollution Index Method

To explore the potentially harmful effects of investigated metals in exposed fish, using the average pollution index method [37], the following formula is used:(4)$$\mathrm{PI}=\frac{1}{n}\sum_{n=1}^{n}\frac{C_{i}}{S_{i}}$$
where PI is the index of trace metal contamination in fish; n is the number of trace metal contaminants; *C_i_* is the average content of trace metal measured in fish; *S_i_* is the evaluation standard of a certain trace metal contaminant (Table 2); the larger the PI value, the more serious the fish is contaminated [38]. According to the level of trace metal content in fish, the PI < 0.1, unpolluted; 0.1–0.2, micro pollution; 0.2–0.5, lightly polluted moderately polluted; 0.5–0.7, moderately polluted; 0.7–1.0, heavily pollution; >1.0, seriously polluted.

### 2.7. Statistical Analysis

Statistical analysis was performed using SPSS (version 26.0, SPSS Inc., Chicago, IL, USA). All statistical analyses to meet assumptions of normality were based on the Kolmogorov–Smirnov test. The Tukey–Kramer method, along with one-factor ANOVA, was applied to detect the differences in heavy metal levels among species’ feeding habits. Linear regression models were used to examine relationships of concentrations of eight trace metals vs. stable isotopic ratios (δ^15^N and δ^13^C). The correlations analysis performed using the Pearson test was used to study the relationship between eight trace metals in the back-muscle of fish with biometric data (length and weight). Results were considered significant at *p* < 0.01 or <0.05 (two-tailed). 

## 3. Results and Discussion

### 3.1. Characteristics of Fish Metal Content

The trace metal contents in the 10 species of fish in Wulungu Lake are listed in Table 2. For different species of fish, the contents of Zn (124 mg kg^−1^) in back-muscle tissue of P.ch, Cu (5.91 mg kg^−1^) and Cd (0.03 mg kg^−1^) in P.sm, Pb (1.09 mg kg^−1^) in C.an, and Ni (2.59 mg kg^−1^) and Cr (1.85 mg kg^−1^) in C.rp were the highest. In general, the trace metal contents in the back-muscle tissue of P. es, B. ss, P. ch, and C. rp were relatively high, while those in the back-muscle tissue of C.an, B.ad, and G.an were relatively low.

Comparing the contamination limits of trace metals in the back-muscle tissue of the fish and the permissible trace metal concentrations in food (Table 2), the average content of only Ni was found to be higher than the contamination limit for all the fish muscles. The As content in muscles of P.es, P.ch, C.an, and S.ca; average Pb content in the muscles of C.an; and average Zn content in the muscles of all fish, except C.an and B.am, were higher than the limit. 

#### Trace Metal Contents in the Muscles of Fish with Different Diets 

Different feeding habits lead to different trace metal contents in organisms [42]. The trace metal contents in muscle tissues of fish with different food habits showed varying trends (Figure 2). The average Hg content in herbivorous fish was minimal, followed by omnivores, carnivores, and planktivores. The biomagnification of Hg in food webs has been reported in various aquatic ecosystems [29,43,44]. It has also been shown that mercury is biomagnified in pelagic fish, but is not effectively transferred in the benthic food chain [45], which is similar to this paper’s result. The average contents of Ni and Cr were found to be in the following order: omnivore > herbivore > planktivore > carnivore. The enrichment degree of Cd in each feeding habit did not differ. The enrichment of Ni, Cr and Cd was not significantly correlated with the feeding level of fish. This result indicates that in addition to feeding habits and nutrient levels, other factors such as fish developmental stages, physiological characteristics, and ecological niches may influence the enrichment characteristics of these trace metals among fish species [46], for Cu, Pb and Zn. The average Cu contents in the muscles of carnivorous and herbivorous fish were the highest. The average Pb contents in the muscles of omnivorous and carnivorous fish were higher compared to the fish in the other two categories of feeding habits. The average Zn content in the muscles of omnivorous fish was much lower than those observed in carnivorous, planktonic, and herbivorous fish. It can be seen that the level of trace metals in carnivorous and omnivorous fish is relatively high, while the level of trace metals in herbivorous fish is low. This may be due to the fact that carnivorous and omnivorous fishes mostly inhabit the lower and middle layers of the water column, while herbivorous fishes are mostly active in the upper and middle layers of the water column. A large amount of trace metals may be sunk in the bottom mud of the water column through migration and transformation. The carnivorous and omnivorous fish may absorb a large amount of contaminated sediment during the feeding process, which leads to a higher degree of trace metal enrichment in their bodies [47,48].

### 3.2. Factors Influencing the Content of Trace Metals in Fish 

#### 3.2.1. Correlation between Trace Metals in Fish and Those in the Water Environment

The quality of the water environment directly affects the level of trace metal content in fish (Table 3). The correlation analysis of the eight types of trace metals in the fish, water body, and sediments of Wulungu Lake (Table 4) revealed that Hg in fish was significantly negatively correlated with Ni in water; Hg in the sediment also showed a significant negative correlation (−0.597, *p* < 0.05; −0.747, *p* < 0.01, respectively). Ni in the fish had a very significant positive (negative) correlation with Hg (As and Ni) in the sediment (Table 4). There was a significant positive correlation between Cu in fish and Ni in water along with a very significant positive correlation with Hg in sediments (0.583, *p* < 0.05; 0.899, *p* < 0.01, respectively). Pb in the fish was significantly negatively correlated with Ni in the water; it was significantly negatively correlated with Hg in the sediment (Table 4). There was a significant negative correlation between Cr in fish and As in water (−0.636, *p* < 0.05). Zn in fish and Ni in the water had a significantly positive correlation with the Hg in sediments (0.585, *p* < 0.05; 0.898, *p* < 0.01, respectively). It can be seen that there is a certain relationship between the trace metals in fish and those in the water environment, and the effect of the trace metal content in sediments on the trace metals in fish bodies is more significant than that of the trace metal content in water. This result can be attributed to the trace metals being non-degradable, easy to accumulate, highly toxic, and enriched through the food chain [49]. As sediments are an integral component of aquatic ecosystems, the degree of trace metal contamination in sediments is indicative of trace metal pollution in these ecosystems [50]. When present in sediments above a certain threshold level, trace metals pose a threat to the ecosystems.

#### 3.2.2. Relationship between Trace Metal Content, Fish Length, and Fish Body Weight

Correlation analysis was carried out on the trace metal content, body length, and fish weight in the lake area (Figure 3). Bodyweight was only significantly negatively correlated with Hg, and body length was significantly positively correlated with As, Cu, Zn, Hg, and significant negative correlation between the contents of Ni, Cr, Pb and Cd. This correlation shows that Hg, Ni, Cr, Pb, Cd levels gradually reduced during the growth of fish, while those of As, Cu, Zn, and Hg increase; this may be a function of the fish metabolism. The younger the individual is, the more active the metabolism; hence, the trace metals are accumulated in higher concentrations in younger individuals. When the fish body length is constant, the amount of muscle tissue in a fish species influences the overall concentration of trace metals; the higher the amount of muscle tissue, the lower the concentration of trace metals [25]. However, as biomagnification occurs along the food chain [20], a correlation can be observed between the content of trace metals and the body length and body weight of the fish. 

The correlation between fish body length and metal concentration in the muscle was investigated. As was significantly positively correlated with Cu, Zn, and Hg; however, significantly negatively correlated with Ni, Cr, Pb, Cd. Ni-Cr, Cr-Pb, and Pb-Cd were significantly positively correlated. These results indicated that As, Cu, Zn, and Hg, and Ni, Cr, Pb, and Cd have the same enrichment characteristics.

### 3.3. Stable Carbon-Nitrogen Isotope Ratios

Carbon and nitrogen stable isotope determination has become an important technical means in studying fish nutritional niches, food web ecology, and in other research fields [51]. Values of δ^15^N and δ^13^C in fish from the study area ranged from 7.37‰ to 15.9‰ and from −27.7‰ to −19.9‰, indicating their wide distribution in the species at different trophic levels (Figure 4A). As is evident from Figure 4B, the δ^13^C in all kinds of food fish collected from Wulungu Lake was in the following order: herbivore > planktivore > carnivore > omnivore, this indicates that the food sources may not be plankton, but instead benthic organisms and plants. Some studies showed that δ^13^C in fish living in the bottom layers of the water body are higher than those in fish living in the upper and middle layers; in submerged plants, δ^13^C is significantly higher than in emergent and floating leaf plants [52]. According to the research [53], the δ^15^N in fish is closely related to their feeding habits and increases with trophic levels. In this study, the δ^15^N in all kinds of food fish collected from Wulungu Lake was in the following order: carnivore > planktivore > omnivore > herbivore. Such results may be due to a large number of nutrients being discharged into the lake [54], affecting the δ^15^N value of the primary food source of the water body, thus increasing the δ^15^N value in the predator fish (at the higher trophic level) [35].

Regression analysis of the relationship between the eight trace metals and carbon and nitrogen isotopes (Figure 5) showed that the concentrations of As, Cu, Cd, and Zn in-creased with the increase in δ^15^N (*p* = 0.000, *p* = 0.000, *p* = 0.03, *p* = 0.007, respectively) and the trophic level. Ni and Cr decreased with the increase in the trophic level (*p* = 0.008, *p* = 0.001, respectively). Hg and Pb showed a downward trend with the increase in the trophic level, but the correlation with δ^15^N was not significant (*p* = 0.705, *p* = 0.431, respectively). It showed that the concentration of As, Cu, Cd and Zn in fish food chain increased with the increase in trophic level, so they would be enriched in fish, and eventually endanger human health through feeding. The concentration of As and Pb increased with the increase in δ^13^C (*p* = 0.003, *p* = 0.004, respectively). The concentrations of Hg, Cu, Cd and Cr increased with the increase in δ^13^C, but the correlation was not significant (*p* = 0.224, *p* = 0.519, *p* = 0.433, *p* = 0.553, respectively), indicating that the pollution of the above metals was due to the consumption of exogenous carbon by fish. The concentrations of Ni and Zn decreased with the increase in δ^13^C, but the correlation of Zn and δ^13^C was not significant (*p* = 0.049, *p* = 0.292, respectively), because it was possible that Ni and Zn mostly appeared as endogenous carbon in fish food.

In summary, the variations in trace metal trends may be due to the presence of the same fish at different trophic levels [55], and the vastness and complexity of the aquatic animal food web. Some scholars have come to different conclusions concerning the trace metal accumulation in organisms at different trophic levels. Yi et al. [4] used carbon and nitrogen isotope technology to study trace metal pollution in fish in the Three Gorges reservoir area and suggested that Hg showed an apparent biological amplification in the food chain; in contrast, in the present study, no such amplification was observed. However, relying solely on carbon and nitrogen isotopes to determine the changing trends and sources of trace metals at different trophic levels may not provide accurate results [56].

### 3.4. Human Health Risk Assessment

Many scholars have carried out relevant studies on the potential health risks of trace metals in fish to consumers. Kumari et al. [57] conducted a health risk assessment of trace metals considering two edible fish in the surface waters of the Jamshedpur urban agglomeration in India. The results showed that risks due to Cr and Pb were higher, and children were more vulnerable than adults.

As shown in Table 5, female EDI calculated for eight trace metals had mean values higher than the male, thus indicating the edible risk of female > male, based on the average daily exposure of the Xinjiang population. The target hazard quotient (*THQ*) values based on gender indicate that the *THQ* of the eight trace metals ingested by fish caught in Wulungu Lake is less than 1, suggesting that there is no health risk of consuming the analyzed fish for eight trace metals intake.

### 3.5. Evaluation of Trace Metal Pollution in Fish

The trace metal contamination index of wild fish in Wulungu Lake is shown in Figure 6, and the contamination index of trace metals in 10 species of wild fish has species differences. The trace metal pollution indices of P.ch and C.rp were both greater than 1, indicating that these two species were seriously polluted by trace metals; G.an, C.an, P.es and S.ca had trace metal pollution indices greater than 0.7 and less than 1, and were in heavily polluted areas. P.sm, B.ss, B.am and B.ad had trace metal pollution indices greater than 0.5 and less than 0.7, and were in heavy pollution and moderate pollution. The trace metal pollution index shows that the investigated metals have high potential harmful effects on fish in Wulungu Lake, which needs our attention.

## 4. Conclusions

This study provides data on the concentrations of As, Cd, Cr, Cu, Hg, Ni, Pb, and Zn in fish samples from Wulungu Lake and an assessment of the health risks that could arise from fish consumption. In the 10 fish species tested, some fish showed higher concentrations of As, Ni, Pb, and Zn, among the trace metals. However, for the population in Xinjiang with an average body weight of 57.3 kg or higher, there was no potential health risk to residents eating fish from Wulungu Lake. The results also show that the *THQ* calculated for both male and female was less than 1, indicating that eating fish from Wulungu Lake in the Xinjiang region does not pose potential health risks. However, the trace metal pollution index shows that the investigated metals have high potential harmful effects on fish in Wulungu Lake. Therefore, the pollution of trace metals in Wulungu Lake is still worthy of attention.

Ni, Cu, and Zn concentrations in fish bodies were positively correlated with Hg in sediments, and Hg and Ni concentrations in fish bodies were negatively correlated with Hg, As, and Ni in sediments. Cu and Zn in fish were positively correlated with Ni in water, and Hg, Pb, and Cr in fish were negatively correlated with Ni and As in water. trace metals in sediments were more significantly correlated with the abundance of trace metals in fish than that in water.

In Wulungu Lake, a simplified food web was constructed based on the predation relations of 10 representative local fish, and the metals showed different trophic transfer behaviors in the food web. This indicates that the species studied can be considered as bioindicators. The correlation of the content of trace metals in fish shows that As has the same enrichment characteristics as those of Cu, Zn, and Hg; furthermore, Ni, Cr, Pb, and Cd have the same enrichment characteristics.

The order of δ^15^N in fish with different feeding habits was as follows: carnivore > planktivore > omnivore > herbivore. The concentrations of As, Cu, Cd, and Zn were positively correlated with δ^15^N, while Ni and Cr were negatively correlated, and Hg and Pb were not significantly correlated with δ^15^N. The concentrations of As and Pb were positively correlated with δ^13^C; those of Hg, Cu, Cd, Cr, and Zn had no significant correlation with δ^13^C, and Ni concentration was negatively correlated with δ^13^C. 

## Figures and Tables

**Figure 1 ijerph-18-09007-f001:**
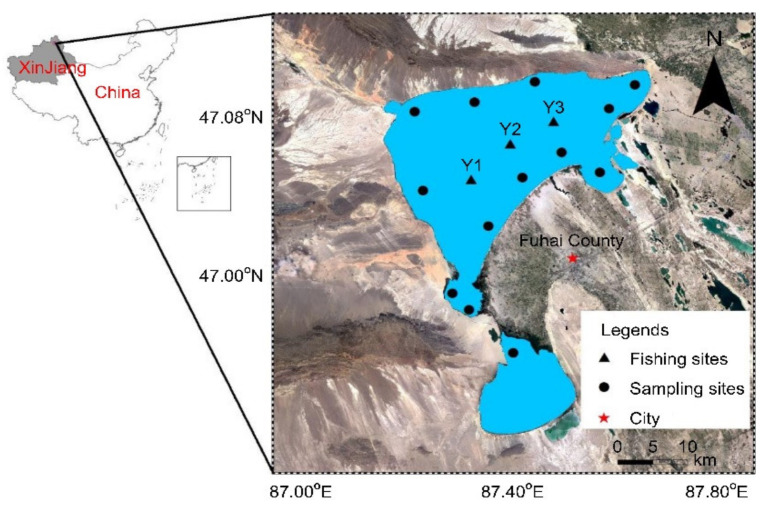
Wulungu Lake and location of the sampling sites.

**Figure 2 ijerph-18-09007-f002:**
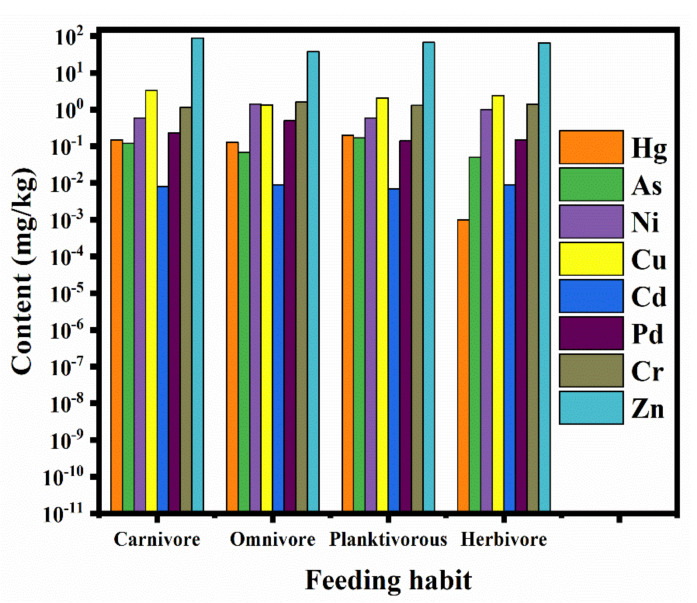
Trace metal contents of fish with different diets.

**Figure 3 ijerph-18-09007-f003:**
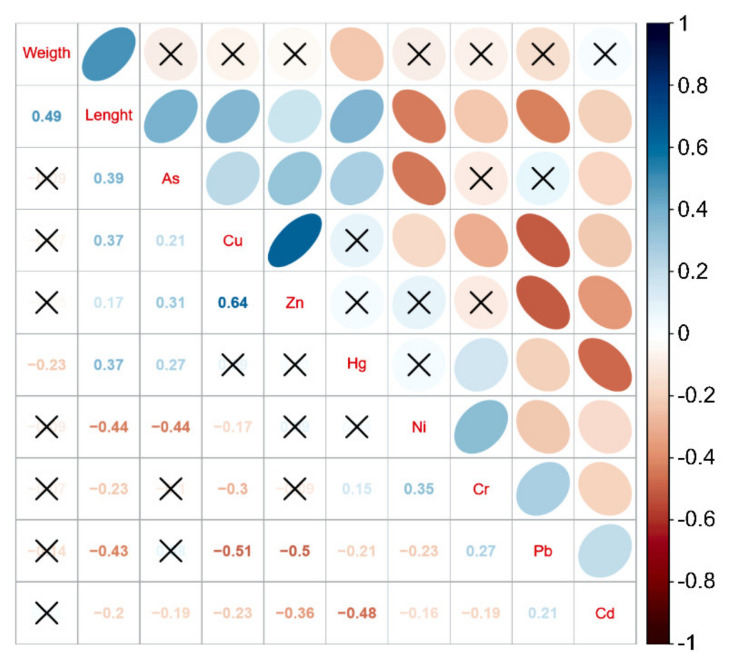
Correlation analysis between fish body length and trace metal. Note: a. The clockwise points indicated a positive correlation and counterclockwise points indicated a negative correlation; b. × implies *p* > 0.05.

**Figure 4 ijerph-18-09007-f004:**
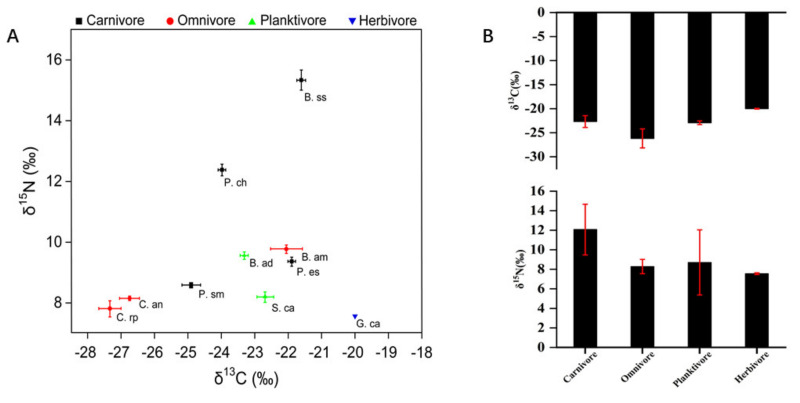
Food web structure (**A**); The comparison of carbon and nitrogen values of fish with varying diets (**B**).

**Figure 5 ijerph-18-09007-f005:**
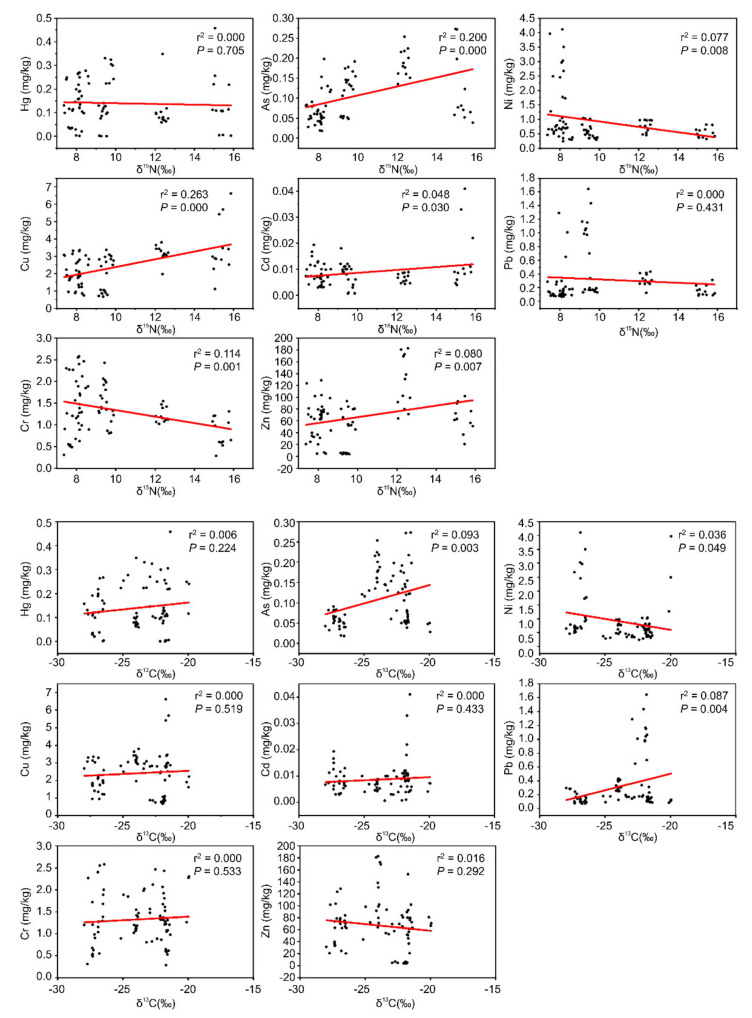
Regression analysis of trace metals and carbon-nitrogen isotopes in fish.

**Figure 6 ijerph-18-09007-f006:**
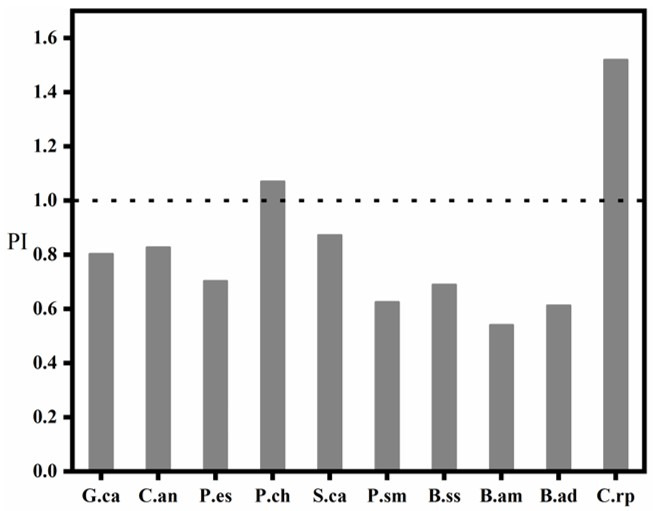
Trace Metal Pollution Index of the fish from Wulungu Lake.

**Table 1 ijerph-18-09007-t001:** Detailed information of the fish captured from Wulungu Lake.

Name ^a^	Sites and Number ^b^	Mean Length (cm)	Mean Weight (g)	Feeding Habit
P.es	Y1(4), Y2(5), Y3(3)	57.48 ± 17.90	1957.20 ± 39.12	Carnivore
B.ss	Y1(6), Y2(5), Y3(1)	31.90 ± 4.60	1146.80 ± 234.70	Carnivore
P.ch	Y1(5), Y2(2), Y3(5)	36.50 ± 2.60	856.60 ± 156.20	Carnivore
P.sm ^c^	-	-	-	Carnivore
C.an	Y1(6), Y2(3), Y3(3)	23.90 ± 1.00	246.10 ± 14.70	Omnivore
C.rp	Y1(8), Y3(4), Y3(0)	25.40 ± 1.80	378.00 ± 57.40	Omnivore
B.am	Y1(0), Y2(0), Y3(5)	39.20 ± 5.4	4901.50 ± 556.60	Omnivore
S.ca	Y1(2), Y2(1), Y3(2)	52.90 ± 4.30	3018.90 ± 943.90	Planktivore
B.ad	Y1(0), Y2(1), Y3(2)	62.50 ± 10.20	34,408.30 ± 18,880.50	Planktivore
G.ca	Y1(1), Y2(2), Y3(0)	38.90 ± 3.20	681.10 ± 13.70	Herbivore

Note: ^a^. P.es (Pikes), B.ss (Bass), P.ch (Perch), P.sm (Pond smelt), C.an (Crucian), C.rp (Carp), B.am (Bream), S.ca (Silver carp), B.ad (Bighead), G.ca (Grass carp). ^b^. Number of test samples for each fish caught, Trace metals, and carbon-nitrogen isotopes. ^c^. Pond smelt are too small, the whole piece is mashed to measure the trace metals content.

**Table 2 ijerph-18-09007-t002:** Average content of trace metals in the back-muscle tissue different eating habits of fish.

Species Feeding Habit		Hg	As	Ni	Cu	Cd	Pb	Cr	Zn
					(mg·kg^−1^)	(*w*/*w*)			
P.es	AM ± SD Range	0.27 ± 0.04 0.22–0.33	0.14 ± 0.02 0.12–0.19	0.35 ± 0.06 0.24–0.46	2.82 ± 0.20 2.49–3.09	0.01 ± 0.00 0.00–0.01	0.16 ± 0.03 0.09–0.19	1.35 ± 0.48 0.81–2.03	68.2 ± 17.9 43.7–98.7
B.ss	AM ± SD Range	0.12 ± 0.02 0.10–0.16	0.06 ± 0.02 0.03–0.08	0.66 ± 0.11 0.46–0.82	3.11 ± 0.25 2.69–3.47	0.01 ± 0.00 0.006–0.01	0.22 ± 0.06 0.14–0.31	0.94 ± 0.54 0.28–2.27	70.1 ± 38.9 21.0–128.6
P.ch	AM ± SD Range	0.08 ± 0.02 0.05–0.12	0.18 ± 0.03 0.17–0.20	0.80 ± 0.18 0.47–0.98	3.28 ± 0.30 2.92–3.80	0.01 ± 0.00 0.004–0.01	0.33 ± 0.07 0.25–0.44	1.25 ± 0.18 1.02–1.55	124 ± 44.1 64.1–183
P.sm	AM ± SD Range	–	0.08 ± 0.04 0.17–0.20	0.37 ± 0.05 0.32–0.42	5.91 ± 0.63 5.42–6.61	0.03 ± 0.01 0.02–0.04	0.10 ± 0.01 0.09–0.12	0.62 ± 0.02 0.61–0.65	102 ± 50.8 51.2–153
C.an	AM ± SD Range	0.11 ± 0.02 0.08–0.15	0.11 ± 0.05 0.05–0.20	0.60 ± 0.15 0.36–0.83	0.86 ± 0.13 0.67–1.06	0.01 ± 0.00 0.007–0.01	1.09 ± 0.27 0.65–1.65	1.77 ± 0.47 0.86–2.47	5.20 ± 1.00 3.77–6.69
C.rp	AM ± SD Range	0.20 ± 0.05 0.12–0.27	0.04 ± 0.01 0.02–0.06	2.59 ± 1.00 1.04–4.11	1.76 ± 0.35 1.20–2.22	0.005 ± 0.0 –	0.10 ± 0.02 0.06–0.13	1.85 ± 0.57 1.04–2.58	72.9 ± 6.76 63.0–84.1
B.am	AM ± SD Range	0.03 ± 0.00 0.03–0.38	0.08 ± 0.01 0.05–0.10	0.63 ± 0.10 0.49–0.75	1.43 ± 0.45 0.93–1.83	0.02 ± 0.00 0.01–0.02	0.09 ± 0.01 0.07–0.09	0.55 ± 0.07 0.49–0.68	33.1 ± 5.85 33.1–40.1
S.ca	AM ± SD Range	0.30 ± 0.10 0.21–0.46	0.24 ± 0.03 0.19–0.27	0.44 ± 0.06 0.36–0.53	2.14 ± 0.67 1.10–2.87	0.006 ± 0.0 0.004–0.01	0.11 ± 0.03 0.08–0.17	1.17 ± 0.25 0.97–1.31	89.1 ± 9.65 63.11–89.5
B.ad	AM ± SD Range	0.01 ± 0.01 0.001–0.02	0.06 ± 0.01 0.04–0.07	0.85 ± 0.15 0.68–0.96	2.00 ± 0.59 1.41–2.60	0.02 ± 0.00 0.01–0.02	0.01 ± 0.01 0.003–0.02	1.46 ± 0.51 0.98–2.00	54.8 ± 30.6 20.4–78.8
G.ca	AM ± SD Range	0.002 ± 0.0 –	0.05 ± 0.0 0.04–0.06	1.03 ± 0.01 1.01–1.04	2.42 ± 0.40 2.01–2.82	0.01 ± 0.01 0.001–0.01	0.16 ± 0.02 0.12–0.17	1.40 ± 0.12 1.27–1.52	66.0 ± 14.3 51.8–80.5
Contamination limits Reference standard		0.50–1 ^a^ [39]	0.10 [39]	0.30 [39]	50 [40]	0.10 [39]	0.50 [39]	2.00 [39]	50 [41]

^a^ Legal limit for methylmercury: 0.5 μg^−1^ WW for fish excluding predatory fish, and 1.0 μg g^−1^ WW for predatory fish.

**Table 3 ijerph-18-09007-t003:** Average content of trace metals of water environment.

Samples		Hg	As	Ni	Cu	Cd	Pb	Cr	Zn
**Water (μg·L^−1^)** **Sediment (mg·kg^−1^)**	**AM ± SD** **Range** **AM ± SD** **Range**	0.09 ± 0.03	0.96 ± 0.60	0.46 ± 0.39	0.13 ± 0.08	0.01 ± 0.01	0.35 ± 0.52	0.19 ± 0.13	2.06 ± 1.12
		0.05–0.19	0.31–2.85	0.15–1.71	0.01–0.32	0.00–0.05	0.09–2.06	0.05–0.54	0.72–4.86
		0.17 ± 0.33	2.76 ± 2.77	12.4 ± 9.37	10.5 ± 10.4	0.07 ± 0.12	13.2 ± 7.81	27.3 ± 16.7	59.8 ± 36.7
		0.00–1.08	0.27–11.2	2.25–34.6	0.25–34.6	0.00–0.48	6.50–35.6	7.24–62.0	27.5–149

**Table 4 ijerph-18-09007-t004:** Correlation between HMs in fish and water environments.

Name	F.Hg	F.As	F.Ni	F.Cu	F.Cd	F.Pb	F.Cr	F.Zn
W.Hg	−0.030	−0.033	−0.012	−0.001	−0.237	0.032	−0.025	0.000
W.As	−0.085	0.157	0.184	−0.051	0.086	0.239	−0.636 *	−0.017
W.Ni	−0.597 *	−0.324	0.477	0.583 *	−0.082	−0.585 *	−0.216	0.585 *
W.Cu	0.182	0.330	−0.165	−0.168	0.222	−0.017	0.448	−0.101
W.Cd	−0.339	0.066	0.296	0.294	−0.079	−0.335	−0.094	0.327
W.Pb	0.162	0.506	−0.139	−0.222	0.138	0.038	0.251	−0.134
W.Cr	0.382	0.292	−0.471	−0.159	−0.097	0.247	0.229	−0.234
W.Zn	0.061	0.346	−0.057	−0.152	0.233	−0.003	0.335	−0.058
S.Hg	−0.747 **	−0.415	0.686 **	0.899 **	0.468	−0.725 **	−0.328	0.898 **
S.As	0.444	−0.198	−0.589 *	−0.353	0.222	0.075	−0.070	−0.334
S.Ni	0.358	−0.262	−0.569 *	−0.381	0.074	0.109	−0.164	−0.419
S.Cu	0.345	−0.124	−0.531	−0.395	0.165	0.200	−0.129	−0.458
S.Cd	0.056	0.012	−0.156	−0.178	0.026	0.114	−0.114	−0.264
S.Pb	0.114	−0.083	−0.246	−0.275	−0.071	0.161	−0.255	−0.346
S.Cr	0.308	0.094	−0.429	−0.287	−0.025	0.242	−0.241	−0.392
S.Zn	0.298	−0.097	−0.392	−0.388	0.127	0.274	−0.258	−0.451

Note: * significant level of correlation is 0.01 (double tailed test); ** significant level of correlation is 0.05 (double tailed test). F, W, and S represent fish, water, and sediments, respectively.

**Table 5 ijerph-18-09007-t005:** *THQ* for males and females of the fish species considered in this study.

Trace Metals	Male					Female			
	RfD (μg/(kg·d))	EDI (μg/(kg·d))		*THQ*		EDI (μg/(kg·d))		*THQ*	
Hg	0.1	0.02	(0.00–0.05)	0.15	(0.00–0.50)	0.02	(0.00–0.06)	0.20	(0.00–0.60)
As	0.3	0.01	(0.00–0.03)	0.04	(0.00–0.10)	0.01	(0.00–0.03)	0.03	(0.00–0.10)
Ni	20	0.10	(0.03–0.44)	0.00	(0.00–0.02)	0.11	(0.03–0.51)	0.01	(0.00–0.03)
Cu	1	0.26	(0.07–0.71)	0.26	(0.07–0.26)	0.3	(0.08–0.82)	0.30	(0.08–0.82)
Cd	1	0.00	–	–	–	–	–	–	–
Pb	4	0.03	(0.01–0.18)	0.01	(0.00–0.05)	0.04	(0.01–0.20)	0.01	(0.00–0.05)
Cr	1500	0.14	(0.03–0.28)	0.00	–	0.16	(0.03–0.32)	0.00	–
Zn	300	7.20	(0.40–19.6)	0.02	(0.00–0.07)	8.31	(0.47–22.7)	0.03	(0.00–0.08)

## Data Availability

The data presented in this study are available on request from the corresponding author.

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
