# Peer review of "Trace Metals (As, Cd, Cr, Cu, Hg, Ni, Pb, Zn) and Stable Isotope Ratios (δ13C and δ15N) in Fish from Wulungu Lake, Xinjiang, China"

_ijerph, 2021, doi:10.3390/ijerph18179007_

Round 1
Reviewer 1 Report
This study uses statistical techniques and correlation studies to holistically explore the content, Spatio-temporal distribution, sources, and potential ecological risks of trace metals. The experimental design was well planned and executed. However, there are few areas that needs some grammar corrections and/or editorial inputs.
Line 45, 49, 90, 101, 110, 136-137, 141, 142, 144, 164, 173, 292, 299, 333, 406-409
Author Response
Dear Reviewer:
Thank you for your comments concerning our manuscript entitled “Trace metals(As, Cd, Cr, Cu, Hg, Ni, Pb, Zn)and stable isotope ratios (δ13C and δ15N) in fish from Wulungu Lake, Xinjiang, China”. Those comments are all valuable and very helpful for revising and improving our paper, as well as the important guiding significance to our researches.
We have tried our best to revise the manuscript and made great changes in the manuscript according to the Reviewers′ comments. Please see the attachment. We look forward to your information about my revised papers and thank you for your good comments.
Yours sincerely
Jianjiang Lu, Prof.
School of Chemistry and Chemical Engineering
Shihezi University
Shihezi, 832003, China
E-mail: lujianjiang2015@163.com
Reviewer 2 Report
We congratulate the authors present a good job, we pointed out some suggestions below.
1, It is suggested to add the trace metal concentrations with previous records in Lake Wulungu
Line 44,…. quantities of trace metal pollutants into Wulungu Lake [5].
2, The conclusions do not indicate whether the species studied can be considered as bioindicators
Line 113, …… a bioindicator for trace metals presence and pollution in lakes in arid regions
3, the authors do not discuss the possible origin of the metals studied
Line 125,… including agricultural wastewater, domestic sewage, and industrial pollution wastewater
Line 346,… .. pollution caused by runoff or human activities.
4, in the evaluation of potential risk vs. average body weight, they do not comment on the risk for people weighing less than 57 kg, such as children.
Author Response
Dear Reviewer:
Thank you for your comments concerning our manuscript entitled “Trace metals(As, Cd, Cr, Cu, Hg, Ni, Pb, Zn)and stable isotope ratios (δ13C and δ15N) in fish from Wulungu Lake, Xinjiang, China”. Those comments are all valuable and very helpful for revising and improving our paper, as well as the important guiding significance to our researches.
We have tried our best to revise the manuscript and made great changes in the manuscript according to the Reviewers′ comments. Please see the attachment. We look forward to your information about my revised papers and thank you for your good comments.
Yours sincerely
Jianjiang Lu, Prof.
School of Chemistry and Chemical Engineering
Shihezi University
Shihezi, 832003, China
E-mail: lujianjiang2015@163.com
Response to Reviewer 2 Comments
Point 1: It is suggested to add the trace metal concentrations with previous records in Lake Wulungu Line 44,…. quantities of trace metal pollutants into Wulungu Lake [5].
Response 1: Thanks for reviewers’ suggestions. We added comments to the revised manuscript (marked up part in page 1, 2 of 19).
Point 2: The conclusions do not indicate whether the species studied can be considered as bioindicators
Line 113, …… a bioindicator for trace metals presence and pollution in lakes in arid regions
Response 2: Thanks for reviewers’ suggestions. We collected 118 common representative edible fish of 4 feeding habits and 10 species in Wulungu Lake. A simplified food web was constructed according to the predator-prey relationship of these fish. The metals showed different nutrient transfer behaviours in the food web, indicating that the species studied can be used as biological indicators. The conclusion is also modified (marked up part in page 13 of 19).
Point 3: the authors do not discuss the possible origin of the metals studied Line 125,… including agricultural wastewater, domestic sewage, and industrial pollution wastewater Line 346,… .. pollution caused by runoff or human activities.
Response 3: The reviewers’ suggestions are valuable for improving our manuscript. In this article, we only discussed whether heavy metals may come from endogenous carbon or exogenous carbon. We did not specifically discuss whether irrigation wastewater, municipal wastewater and industrial pollution wastewater contribute, but only quoted the previous conclusions. Sources of heavy metals are published in another article. We apologize for your misunderstanding due to our improper description, so we have made some modifications (marked up part in page 10 of 19).
Point 4: in the evaluation of potential risk vs. average body weight, they do not comment on the risk for people weighing less than 57 kg, such as children.
Response 4: Thanks for reviewers’ valuable suggestions. EDI (estimated daily intake calculation) calculation needed data of the daily intake and average body weight of aquatic products. We searched the relevant literature and did not find the daily intake of aquatic products of children in Xinjiang. As long as the statistical data of adult men and women and daily intake of aquatic products, the average body weight of adult women in Xinjiang is 57.3kg. Through THQ calculation, there is no potential health risk if the weight exceeds 57.3kg.

Reviewer 3 Report
The authors of the paper entitled: "Trace metals(As, Cd, Cr, Cu, Hg, Ni, Pb, Zn)and stable isotope ratios (δ13C and δ15N) in fish from Wulungu Lake, Xinjiang, China" aimed to evaluate the exposure levels of different fish species living in Wulungu lake to trace metals.
The novelty of the paper is fair, the introduction is well written, the methodological approach is pertinent but the results are very difficult to understand. Overall, the manuscript should be simplified and clarified. Moreover, the discussion part should be implemented.
Here are the main concerns:
- Line 61-62: please add the following references in the introduction part (Zezza, D., Bisegna, A., Angelozzi, G., Merola, C., Conte, A., Amorena, M., & Perugini, M. (2020). Impact of endocrine disruptors on vitellogenin concentrations in wild brown trout (Salmo trutta trutta). Bulletin of Environmental Contamination and Toxicology, 105(2), 218-223.) and (Merola, C., Bisegna, A., Angelozzi, G., Conte, A., Abete, M. C., Stella, C., ... & Perugini, M. (2021). Study of Heavy Metals Pollution and Vitellogenin Levels in Brown Trout (Salmo trutta trutta) Wild Fish Populations. Applied Sciences, 11(11), 4965.)
- Line 141: please clarify the reference
- Table 2 is complex and the species of the fish are not categorized correctly according to feeding habitat. I suggest splitting the table, separating the results of fish muscle with those of sediments or water.
- Line 122 (3.1.1. Trace metal contents in the muscles of fish with different diets): please discuss the results according to published papers available in the literature.
- Line 155: please clarify this sentence, in which way younger animals could have higher concentrations of heavy metals compared to the older ones? Moreover, add references related to this statement.
- Please describe the potentially harmful effects of investigated metals in exposed fish.
- In which way the authors expressed the concentrations of metals? (e.g. wet weight).
- Why the authors did not consider the influence of the sampling site on metals concentrations determined in exposed fish?
Author Response
Dear Reviewer:
Thank you for your comments concerning our manuscript entitled “Trace metals(As, Cd, Cr, Cu, Hg, Ni, Pb, Zn)and stable isotope ratios (δ13C and δ15N) in fish from Wulungu Lake, Xinjiang, China”. Those comments are all valuable and very helpful for revising and improving our paper, as well as the important guiding significance to our researches.
We have tried our best to revise the manuscript and made great changes in the manuscript according to the Reviewers′ comments. Please see the attachment. We look forward to your information about my revised papers and thank you for your good comments.
Yours sincerely
Jianjiang Lu, Prof.
School of Chemistry and Chemical Engineering
Shihezi University
Shihezi, 832003, China
E-mail: lujianjiang2015@163.com
Response to Reviewer 2 Comments
Point 1:Line 61-62: please add the following references in the introduction part (Zezza, D., Bisegna, A., Angelozzi, G., Merola, C., Conte, A., Amorena, M., & Perugini, M. (2020). Impact of endocrine disruptors on vitellogenin concentrations in wild brown trout (Salmo trutta trutta). Bulletin of Environmental Contamination and Toxicology, 105(2), 218-223.) and (Merola, C., Bisegna, A., Angelozzi, G., Conte, A., Abete, M. C., Stella, C., ... & Perugini, M. (2021). Study of Trace Metals Pollution and Vitellogenin Levels in Brown Trout (Salmo trutta trutta) Wild Fish Populations. Applied Sciences, 11(11), 4965. )
Response 1: Thanks for reviewers’ suggestions. We read the two articles you listed and decided to add references on the bioaccumulation of trace metals in fish in the introduction section((Zezza, D et al., 2020; Merola, C et al., 2021) (refs. 15 and 16).
Point 2:Line 141: please clarify the reference
Response 2: Thanks for your suggestions. We have changed the citation format of this standard according to the requirements of this journal (refs. 27 and 28).
Point 3:Table 2 is complex and the species of the fish are not categorized correctly according to feeding habitat. I suggest splitting the table, separating the results of fish muscle with those of sediments or water.
Response 3: Thank you for your advice. We splitted the table, separating the results of fish muscle with those of sediments or water. We determined that fish species are correctly classified according to feeding habitat. Carnivore has four species, Omnivore has three species, Planktivore has two species, and Herbivore has one species(marked up part in page 6 of 19)..
Point 4: Line 122 (3.1.1. Trace metal contents in the muscles of fish with different diets): please discuss the results according to published papers available in the literature.
Response 4: Thanks for your suggestions. This part has been rewritten and the figure has been added (marked up part in page 7 of 19).
Point 5: Line 155: please clarify this sentence, in which way younger animals could have higher concentrations of trace metals compared to the older ones? Moreover, add references related to this statement.
Response 5: Thank you for pointing out this problem.
From the physiological and biochemical point of view, the younger the individual, the more vigorous the metabolic activity, and the older the individual, the weaker the metabolism, so the younger the individual, the higher the amount of metal elements, but also may be with the accumulation of trace metal elements to a certain extent, the body's absorption is reduced or excretion is enhanced or both(Douben PE. Lead and cadmium in stone loach (Noemacheilus barbatulus L.) from three rivers in Derbyshire. Ecotoxicol Environ Saf. 1989 Aug;18(1):35-58. doi: 10.1016/0147-6513(89)90090-0. PMID: 2776688 ;Widianarko B, Van Gestel CA, Verweij RA, Van Straalen NM. Associations between trace metals in sediment, water, and guppy, Poecilia reticulata (Peters), from urban streams of Semarang, Indonesia. Ecotoxicol Environ Saf. 2000 May;46(1):101-7. doi: 10.1006/eesa.1999.1879. PMID: 10806000;). The total content of trace metals in fish will increase with the increase of body weight, and the stage of the fastest growth of fish length may also be the fastest accumulation of trace metals, when entering the stage of slow growth of body length and rapid increase of body weight, the rapid growth of body weight has a dilution effect on fish weight metals (Liu P, Zhou YQ, Zang LJ. [Investigation of trace metal contamination in four kinds of fishes from the different farmer markets in Beijing]. Huan Jing Ke Xue. 2011 Jul;32(7):2062-8. Chinese. PMID: 21922831).
PETER E. T. DOUBEN concluded that fish Noemacheilus barbatulus L. (stone loach) were caught at about 4-week intervals from single sites in three Derbyshire rivers, with different concentrations of cadmium and lead in sediments and water, during a l-year sampling program,A steady state of cadmium burden was reached by fish of 2 years old or more but not by younger fish.
B Widianarko et al found that Significant differences were found in metal body concentrations (Pb and Zn) between fish collected from sites with different degrees of pollution. A significant declining trend of Pb concentrations with increasing organism size was observed.
Point 6:Please describe the potentially harmful effects of investigated metals in exposed fish.
Response 6: Thank you for pointing out this problem. In the revised draft, we added sections 2.6 (marked up part in page 5 of 19) and 3.5 (marked up part in page 12 of 19) to illustrate this problem.
Point 7: In which way the authors expressed the concentrations of metals? (e.g. wet weight).
Response 7: Thank you for pointing out this problem. The metal concentrations (in WW) in the 10 commercial species (a total of 118 fish ) in Table 2.
Point 8:Why the authors did not consider the influence of the sampling site on metals concentrations determined in exposed fish?
Response 8: Thank you for your advice. We considered the influence of the sampling site on metals concentrations determined in exposed fish. The influence of the sampling site on metals concentrations was found to be negligible after we obtained trace metal content data. At 3 fishing locations (Y1, Y2, Y3) in Wulungu Lake, every 2 to 5 km. We speculate that the distance of 2 to 5 km is too short to have much effect on the enrichment of trace metals in fish. We will seriously consider your suggestions and implement them in our later work.

Round 2
Reviewer 3 Report
The Authors improved the quality of the manuscript according to the referee's suggestions, so it is ready for publication in the present form.
This manuscript is a resubmission of an earlier submission. The following is a list of the peer review reports and author responses from that submission.